

# Anonymous group structure algorithm based on community structure

Linghong Kuang, Kunliang Si and Jing Zhang

School of Computer Science and Mathematics, Fujian University of Technology, Fuzhou, Fujian, China

## ABSTRACT

A social network is a platform that users can share data through the internet. With the ever-increasing intertwining of social networks and daily existence, the accumulation of personal privacy information is steadily mounting. However, the exposure of such data could lead to disastrous consequences. To mitigate this problem, an anonymous group structure algorithm based on community structure is proposed in this article. At first, a privacy protection scheme model is designed, which can be adjusted dynamically according to the network size and user demand. Secondly, based on the community characteristics, the concept of fuzzy subordinate degree is introduced, then three kinds of community structure mining algorithms are designed: the fuzzy subordinate degree-based algorithm, the improved Kernighan-Lin algorithm, and the enhanced label propagation algorithm. At last, according to the level of privacy, different anonymous graph construction algorithms based on community structure are designed. Furthermore, the simulation experiments show that the three methods of community division can divide the network community effectively. They can be utilized at different privacy levels. In addition, the scheme can satisfy the privacy requirement with minor changes.

## INTRODUCTION

Convenient and rapid smart services, including locating nearby hospitals, banks, and dining options, are accessed by individuals through social platforms. Massive data sharing improves service quality but also leads to multiple privacy breaches (*Li et al., 2020*; *Kong et al., 2024*). Sharing data while ensuring privacy and security has emerged as a paramount concern across various industries. Through the analysis of the social network data, service providers can glean insights such as user preferences to support the further services. However, the convenience offered by social networks has given rise to a series of challenges and problems. The issues related to the security of personal privacy information are taking on a more pronounced role.

Most current protection solutions utilize anonymity technology for data release, which involves the removal of identity information such as user names from the data. However, attackers can easily analyze a plethora of potential information through network topology. As illustrated in Fig. 1, the labeled nodes represent users, and the connection lines indicate mutual acquaintance between the pairs. When an attacker discovers that an individual

Corresponding author
Jing Zhang, jing165455@126.com

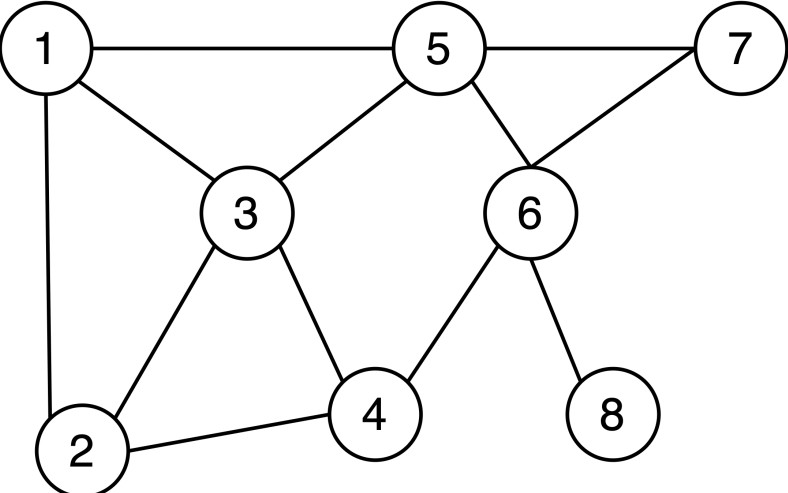

**Figure 1** **Example of social network topology.**

has four friends and two of those friends are acquainted with each other, even without the user's name in the depiction, it becomes readily apparent that node 6 is the target the attacker should focus on. Therefore, there is an urgent need to devise a viable solution to safeguard the shared network topology.

In social networks, some users are closely connected due to various factors. These users are referred to as the community structure in the graph. Considering that the community structure in graph data remains stable and holds a vast amount of potential information, leveraging this structure for data privacy protection makes eminent sense. However, research on privacy protection for shared data still faces numerous challenges: how to design a privacy protection scheme that balances data validity and privacy security; how to design a scheme that protects users based on their privacy needs; and how to mitigate structural feature leakage. To tackle these challenging problems, this article presents a privacy-preserving scheme based on multiple anonymizing graph structure, the main contributions of this article are as follows:

(1) An anonymization protection platform has been implemented, offering users the flexibility to select suitable community mining algorithms and privacy protection algorithms based on varying network sizes, security requirements, and data permissions.

(2) A fuzzy subordinate degree-based community structure mining algorithm is designed. In this article, the concept of a fuzzy subordinate degree in community mining is introduced and utilized as the foundation for community division. Three community structure mining algorithms are designed: the fuzzy subordinate degree-based algorithm, the improved Kernighan-Lin algorithm, and the enhanced label propagation algorithm. These algorithms aim to comprehensively extract the potential association structure of the social network and establish the association structure model.

(3) Design a privacy protection scheme based on anonymous graphs. Based on the division of community structure, three privacy protection schemes are designed according to different levels of privacy protection.

# RELATED WORKS

Privacy-preserving schemes can be broadly categorized according to their design mechanisms, including cryptography, anonymous graph construction, and differential privacy-preserving techniques (*Dwork et al., 2006*). In recent years, interdisciplinary privacy protection schemes based on anonymous graphs have attracted the attention of numerous scholars in various fields, including computer science and mathematics (*Wang et al., 2018*).

## K-anonymity

In the anonymous protection scheme, utilizing random numbers or letters instead of node IDs represents a relatively simple and direct protection approach (*Cai et al., 2016*). The second method involves anonymizing graph data by either adding virtual nodes or deleting existing nodes (*Siddula et al., 2019*). The third category achieves anonymization of graph data by adding virtual edges or deleting existing edges (*Ji, Mittal & Beyah, 2016*). There are various methods to achieve edge anonymization, such as utilizing random walks. After selecting the initial node, the process involves random traversal, and the resulting path represents the newly added edges. Structural similarity is regarded as the optimal solution for anonymizing graph data (*Siddula et al., 2019*). It achieves K-anonymity by constructing a structure with high similarity. K-anonymity is a method in that the individuals who are the subjects of the data cannot be re-identified while the data remain practically (*Ye et al., 2020*). In the earliest days, K-1 nodes were chosen by a random function thus satisfying K-anonymity. In recent years, *Wang et al. (2018)* utilizes adding or removing edges to construct K-anonymitys. *Hao et al., (2024)* propose a multi-level k-degree anonymity (MLDA) scheme on directed social network graphs.

*Campan & Truta (2008)* propose Social Network Greedy Anonymization (SaNGreeA) to anonymize social networks and quantify the information loss caused by edge generalization, which can prevent the leakage of quasi-identifiers; *Wang et al. (2015)* points out that the SaNGreeA algorithm does not protect sensitive attributes. To solve this problem, the Masking Algorithm for Social Networks (MASN) is proposed to protect the user's quasi-identifying attributes and sensitive attributes, but the efficiency of the algorithm is significantly reduced. *Song et al. (2019)* proposed a random K-anonymity algorithm to solve the problem that traditional K-anonymity contains at least k identical records. To make k records indistinguishable, noise is added to numerical attributes, and non-numeric attributes are randomized, which can effectively resist poverty. Lift attack and improve operational efficiency through two-step clustering. However, when the range of numerical attributes is large, the performance decreases. *Nettleton, Torra & Dries (2014)* compare node/edge clustering and node/edge modification methods and tested K-anonymization on online social network type graphs. *Zhang et al., (2021)* propose a privacy protection scheme based on graph matching, which achieved K-anonymity through node clustering and graph modification to protect user privacy. At the same time, degree-based entropy

is introduced to improve the accuracy of node clustering and resist neighbor attacks. However, it is easily affected by the distribution of degree and clustering coefficient. *Gao et al. (2023)* propose a new privacy protection method based on compressed sensing to resist node-level background knowledge attacks. *Yazdanjue et al. (2024)* propose an enhanced discrete particle swarm optimization (EDPSO) algorithm, which effectively minimizes the SIL within the clustering process of the structural k-anonymity model. They first compressed node information, and then randomly deleted and changed link relationships in the form of label groups to blur node degrees. To anonymize social networks, and finally combine the reconstruction algorithm with the measurement matrix to ensure high-precision reconstruction of social networks and ensure the overall efficiency of the method.

## Community structure mining algorithm

Community division is a fundamental task in network analysis and graph mining, as it helps uncover the underlying structure and organization of complex networks. By identifying communities, researchers can gain insights into the functional organization of networks, identify key nodes or groups, and understand the dynamics of information flow or interaction patterns within the network.

*Kernighan & Lin (1970)* propose the Kernighan-Lin (KL) classical network partitioning algorithm, which is a trial-optimized bisection mining method designed according to the greedy principle. *Pothen, Simon & Liou (1990)* develop an algorithm that employs Laplacice matrix eigenvalues as a measure of the quality of community division. The spectral bisection method analyzes an undirected weighted graph abstracted from social networks using spectral graph partitioning theory. Ultimately, it divides the network graph into two subgraphs. In recent years, scholars have developed various community latent structure mining methods, which broadly fall into the above two categories. These include dependency-based algorithms (*Wang & Zhao, 2014*), node similarity-based algorithms (*Saoud & Moussaoui, 2018*), labeling-based algorithms (*Meng & Ling-juan, 2020*). A summary *Casas-Roma, Herrera-Joancomartí & Torra (2017)* provides an analysis of recent years employing graph-theoretic methods for privacy preservation. *Zhou et al. (2023)* propose unsupervised attributed network embedding (CDBNE) to resolve the issue of utilizing clustering-oriented information. *Ji et al. (2023)* construct an overlapping community-driven feedback mechanism for improving consensus in SN-GDM, which improve the accuracy of overlapping community detection. *Das, Devarapalli & Biswas (2024)* propose a novel information diffusion-based approach to leverage the latent information exchange among individuals for identifying communities.

In summary, while some studies have focused on graph data privacy protection, there remain some shortcomings:

(1) The platform predominantly employs a uniform privacy protection scheme, often without considering the level of privacy protection and permission assignment, resulting in limited adaptability to dynamic changes.

(2) Failure to consider the importance of community in privacy protection leads to an increase in the number of nodes and edges that need to be changed.

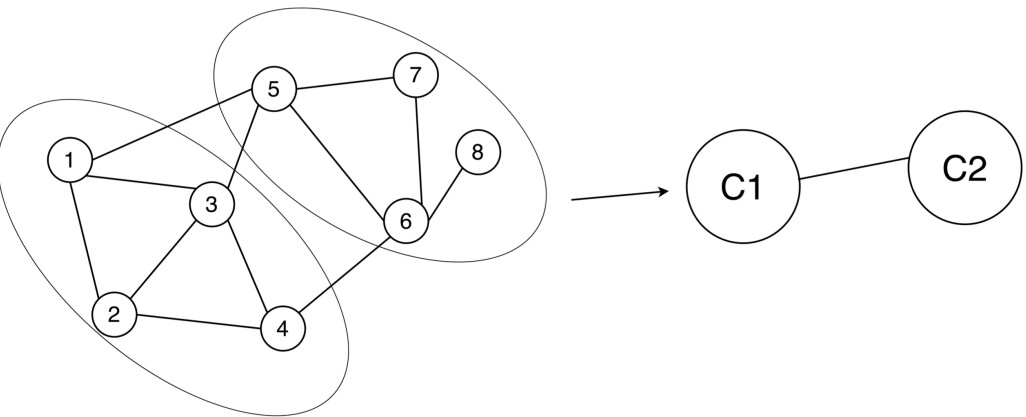

**Figure 2** **Example of network graph model.** If node 1 and node 2 are related, then there is an edge between node 1 and node 2 in the graph. When there is a relationship between nodes, there are edges connecting the nodes. The ellipse represents the divided community structure, and C1 and C2 are anonymous graphs using super nodes to replace the community structure.

(3) The community structure division algorithm does not consider the fuzzy subordinative degree, resulting in division outcomes that significantly diverge from the actual situation.

Therefore, a privacy protection scheme based on community and anonymous graph structures is designed in this article.

## PRELIMINARIES

**Graph** (*Blondel et al., 2008*)**:** Social networks are commonly represented as graph data structures in scientific research. In this representation, network participants are depicted as nodes in the graph, while the connections between participants are represented as edges. This abstraction serves to streamline the analysis and modeling of social networks, enabling researchers to systematically investigate a wide range of phenomena and behaviors within these networks. The scheme proposed in this article focuses on the privacy-preserving publication of undirected graphs $G(V, E)$ in social graph data. The set V represents the nodes, or participants, while the set E represents the edges or connections.

**Community detection** (*Blondel et al., 2008*)**:** Community detection involves the partition of nodes within a network into distinct subsets. These subsets are characterized by nodes that exhibit similar attributes or maintain strong connections among themselves, whereas the connections between nodes belonging to different subsets tend to be comparatively weaker or sparser (see Fig. 2).

**K-anonymity** (*Samarati & Sweeney, 1998*)**:** If for each row $r \in D$, there exist at least $K - 1$ other rows $r_1, \ldots, r_{k-1} \in D$, such $\Pi_{qi(D)} r = \Pi_{qi(D)} r_1, \ldots, \Pi_{qi(D)} r = \Pi_{qi(D)} r_{k-1}$, then a dataset satisfies K-Anonymity. Informally, a dataset is "K-Anonymized" for a particular, if each individual in the dataset is a member of a group of size at least, such that each member of the group shares the same quasi-identifiers (a selected subset of all the dataset's columns) with all other members of the group.

**Fuzzy subordinative degree:** The shortest path length from node $u$ to Community $C$ can be determined. According to Eq. (1), the longer this path length, the smaller the subordinative degree. $N(u)$ represents the set of neighboring nodes of node $u$. If node $u$ is not connected to $C$, the subordinative degree is defined as 0.

$$f(u,C) = \begin{cases} \dfrac{1}{l(u,C)} \displaystyle\sum_{v_i \in N(u) \bigcap N(C)} \dfrac{1}{d_i} & \text{Node u connect C} \\ 0 & \text{Node u not connect C} \end{cases} \quad (1)$$

**Modularity (*Huang et al., 2022*):** Modularity is usually used to evaluate the quality of community division results. A higher module degree represents a better structure of the mined communities, which takes the value in the range of [0, 1], generally when the value of the module degree is in the range of 0.3 to 0.7 it indicates that the mining results are good. The Modularity Q is defined in Eq. (2):

$$Q = \sum_i (e_{ii} - a_i^2) \quad (2)$$

A social network is decomposed into $m$ communities and corresponds to an $m*m$-dimensional symmetric matrix $e$, where element $e_{ij}$ means the proportion of the number of edges connected by the nodes in the community $i$ to the nodes in the community $i$ in all edges, and $e_{ii}$ means the proportion of the internal connections of nodes in community $i$ in all edges. $a_i = \sum_j e_{ij}$ represents the proportion of the number of edges connected to the nodes in the community $i$ in all edges

## METHODS

### Overview of our proposed scheme

This section introduces the privacy protection scheme based on anonymized graphs, as illustrated in Fig. 3. Users will send the shared data to the server, which will then generate different levels of anonymized graphs based on the anonymity requirements. Each user can view the corresponding level of anonymized graphs according to their permissions. The anonymity graph construction algorithm is carried out in two steps:

(1) Fuzzy subordinative degree community structure construction: research and analysis of the topology of different network sizes, and divide the potential structure according to the Fuzzy subordinative degree judgment.

(2) Anonymity graph construction scheme: set the anonymity level and select the corresponding anonymity scheme.

### Community structure mining algorithm

In this article, the concept of fuzzy subordinative degree is introduced, based on which three association structure mining algorithms are designed: The fuzzy subordinate degree-based algorithm, the improved KL algorithm, and the enhanced label propagation algorithm.

#### *Fuzzy subordinative degree community structure mining algorithm*

This algorithm utilizes fuzzy subordinative degree as a measure to determine whether nodes are classified into the same community. In the initial phase of the algorithm, several

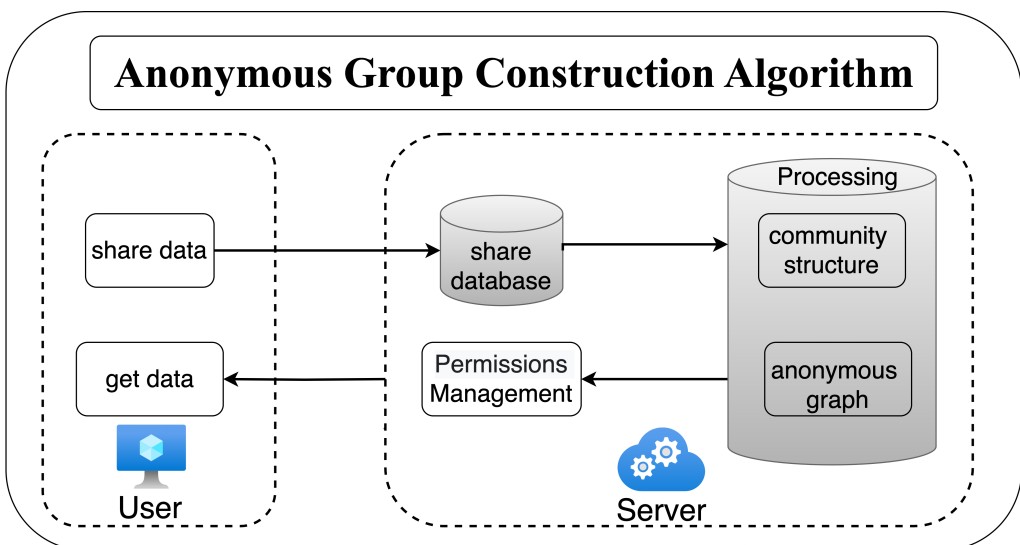

**Figure 3** The flow of privacy protection scheme.

communities are initialized and the affiliation of each node with these communities is calculated using the fuzzy subordinative degree definition formula. Then the communities are divided according to the subordinative degree matrix. To improve the accuracy of community division, it is necessary to set a threshold value of 1 to filter out the nodes whose subordinative degree is too small. Through experimental analysis, it was found that the number of community structures classified based on this method alone was high and the modularity was low. So increase the threshold value 2 to secondary merge the results to adjust those nodes with higher subordinative degree but not divided into the same community, which can get a higher modularity degree.

---

**Algorithm 1** Fuzzy Subordinative degree Community Structure mining Algorithm (FSCA)

---

**Input:** Graph $G$

**Output:** subordinative degree of nodes $f(u_i, C)$, communities $C$

  1:  Select the $c$ nodes with the largest degree as the initial communities

  2:  Set the threshold $\eta_1$.

  3:  calculate the fuzzy subordinative degree.

  4:  **if** fuzzy subordinative degree $> \eta_1$ **then**

  5:      merge communities

  6:  **end if**

  7:  Update subordinative degree

  8:  Join the community with the highest subordinative degree

  9:  **if** subordinative degree $< \eta_2$ **then**

10:      Create a new community

11:  **end if**

---

### *Improved KL community mining algorithm*

KL algorithm is suitable for smaller network sizes. The process is as follows: initially randomize the two communities, then iteratively optimize the communities by exchanging node pairs in two combinations, and decide the community to which each node belongs based on the set gain value. In this article, the gain values are set based on the fuzzy subordinative degree.

If there are two nodes $v_i, v_j$ and where $v_i$ belongs to the community $C_i$ and $v_j$ belongs to the community $C_j$, then the gain value of exchanging the two nodes is defined as Eq. (3). Where $X_i = X^{out} - X^{in}$.

$$g_{ij} = X_i + X_j - 2f_{ij}. \tag{3}$$

If node $v_i$ belongs to community $C_i$, its internal subordinative degree is defined as $X_i^{in} = \sum_{v_k \in C_i} f_{ik}$, where $f_{ik}$ is the fuzzy subordinative degree between two nodes $v_i$ and $v_k$, such as one of the $v_k$ is regarded as a new community, so that the calculation method is the same as Eq. (1); if node $v_i$ belongs to the community $C_j$, its external subordinative degree is defined as $X_i^{out} = \sum_{v_k \in C_j} f_{ik}$.

---

**Algorithm 2** Improved KL community mining Algorithm (KLA)

**Input:** Graph $G$

**Output:** communities $C$

  1: Divide $G$ equally into two communities based on the number of nodes
  2: Calculate the gain value $g$ of the two communities according to Equation 3
  3: exchange two nodes, and exchange the pair with the largest difference in gain value $g$
  4: Update the weight difference between inside and outside the remaining nodes
  5: Repeat 2 and 4 so that all nodes experience an exchange
  6: The sequence of node exchanges with the largest cumulative gain is selected as the result of the division according to the order of node pair exchanges

---

### *Enhanced label propagation community mining algorithm*

This algorithm is designed based on the propagation properties of social networks. In a social network, each node has a label that ideally should be consistent with most of its neighbors. Unlike traditional labeling algorithms, this algorithm assigns different labels to each node in the network based on its subordinative degree, which indicates the community to which the node belongs. It leverages the network propagation characteristics by acting as a receiver in each iteration to receive the new labels obtained. After several rounds of iterations, the algorithm eventually converges, ensuring that nodes with the same label are grouped into the same community. During propagation, the current label is updated to the one with the highest subordinative degree and the highest cumulative number of iterations.

## Anonymous graph construction

The objective of this section is to design privacy anonymization schemes. When structural features are deemed capable of compromising privacy, constructing the anonymizing

graph becomes imperative to safeguard individual privacy. Traditionally, the common practice involves adding or removing nodes or edges from the data. This article aims to design various anonymizing graph construction algorithms tailored to different security requirements. The overarching research objective is to uphold the privacy of each user to prevent the disclosure of structural information about them. To achieve this goal, three schemes with different levels of protection were designed. The setting of privacy protection level depends on the data owner. The first type, generalization is employed to minimize externally obtained information by aggregating all individuals into a larger scope. For instance, a teacher in a school may be generalized and represented as a super node of the school. In the second type, by adjusting the weights of nodes and edges, anonymized graph information can be partially disclosed to the outside, providing more information compared to the generalization scheme. Lastly, without altering the nodes, adjusting the weights of the edges maximizes externally obtained information while still ensuring the protection of the user's information.

---

**Algorithm 3** Enhanced label Propagation Community mining Algorithm (LPA)

---

**Input:** Graph $G$

**Output:** communities $C$

  1: Get the current labels of all nodes and all their neighbor nodes

  2: Calculate the subordinative degree based on Equation 1

  3: Update node labels and their counts based on the subordinative degree

  4: Find the most appearing label

  5: **if** The current node label is in the most labeled set **then**

  6:     start the next round of the loop

  7: **end if**

---

## PERFORMANCE ANALYSIS

In this article, we have designed a simulation system using QT with the features shown in Fig. 4. After importing the data it can be visualized, anonymized and other operations can be performed on the data according to the requirements. In this article, three community mining algorithms are first compared and then the next step of anonymization is done using the better algorithm.

### Data sets and experiment environments

Experiments on three public social network datasets. Datasets can be downloaded from https://networks.skewed.de. The datasets are uniformly processed as undirected unweighted graphs before the experiments. The datasets are described as follows. The details are shown in Table 1.

    Zachary karate club (*Zachary, 1977*): Network of friendships among members of a university karate club.

---

**Algorithm 4** Anonymous Graph Construction Algorithm (AGCA)

---

**Input:** Graph $G$

**Output:** Anonymous Graph $G'$

1: Set anonymity level
2: Choose an anonymizing graph construction method based on the degree of anonymity
3: **if** level equal 1 **then**
4:    Generalization. Each node within the community is replaced by the community to which it belongs
5: **end if**
6: **if** level equal 2 **then**
7:    Both nodes and edges are altered by adding and deleting certain nodes and edges. This process ensures that the association internally forms an L-regular graph, where $L$ represents the average degree within the community. The counts of edges represent the average fuzzy subordinative degree between two nodes. Additionally, edges between communities are added if a connection exists between the communities.
8: **end if**
9: **if** level equal 3 **then**
10:    The set of nodes remains unchanged while the edges are modified. Add or remove the corresponding edges as required, and assign weights to them. The original edge weights are set to range from 0.6 to 1, while the weights of newly added edges are set to range from 0 to 0.5.
11: **end if**

---

Dolphin social network (*Lusseau et al., 2003*): An undirected social network of frequent associations observed among 62 dolphins (Tursiops) in a community living off Doubtful Sound, New Zealand, from 1994–2001.

College football network (*Girvan & Newman, 2002*): A network of American football games between Division IA colleges during regular season Fall 2000.

Experimental environments:

The hardware configuration includes an Intel(R) Core(TM) i3-12100 processor with 24.0GB of memory. The operating system is Windows 10, a 64-bit OS. The program is implemented in Python, utilizing the PyCharm development environment as the programming tool.

## Experiments
### *Community mining*

*Running time.* Table 2 displays the runtime performance of the three algorithms in the three datasets. For a network comprising $n$ nodes and $m$ edges, FSCA primarily focuses on identifying nodes for exchange, with a time complexity of $O(n^2)$. KLA primarily deals with acquiring the fuzzy subordinative degree matrix, with a time complexity of $O(n^2)$. On the other hand, LPA primarily revolves around label propagation, with a time complexity close to linear, $O(n)$.

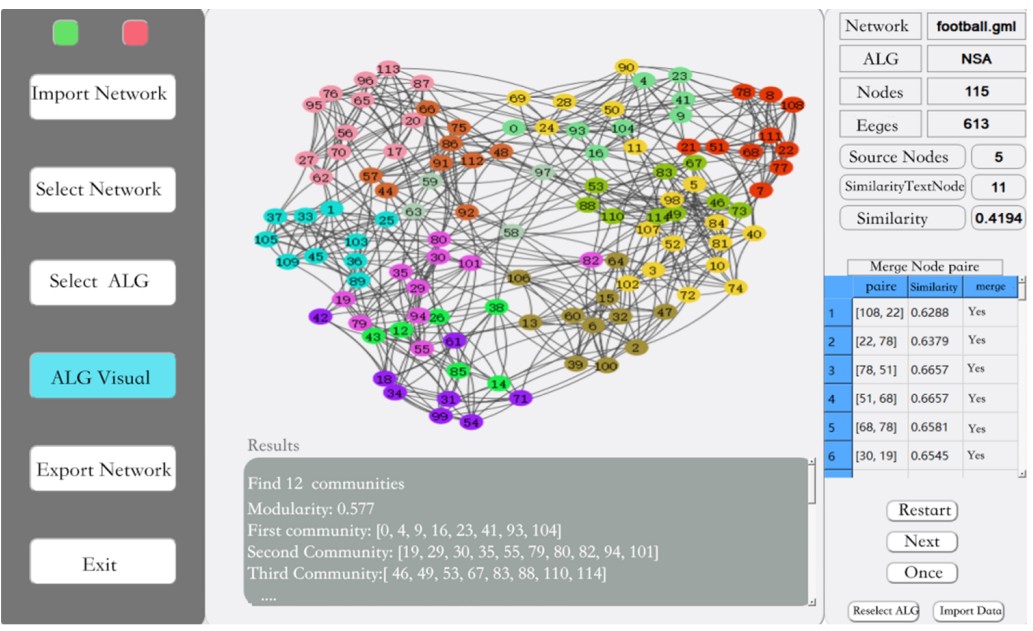

**Figure 4 Simulation system.**

**Table 1 Dataset.**

|          | Nodes | Edges | Clustering-coefficient |
|----------|-------|-------|------------------------|
| Karate   | 34    | 77    | 0.26                   |
| Dolphin  | 62    | 159   | 0.31                   |
| Football | 115   | 613   | 0.41                   |

**Table 2 Comparison of running time.**

|         | Time (ms) | | |
|---------|------|-----|-----|
| Network | FSCA | KLA | LPA |
| Karate   | 8  | 3  | 2  |
| Football | 16 | 10 | 7  |
| Dolphin  | 64 | 27 | 11 |

In terms of runtime, the three algorithms offer insight into the scale of networks to which they are most applicable. For instance, LPA demonstrates relatively higher efficiency compared to the other two algorithms, boasting a faster division speed, rendering it more suitable for mining the data structure of large-scale networks.

## Modularity

From the Table 3, it can be seen that the LPA algorithm has the largest value of modularity, which indicates that LPA is the best among the three community mining algorithms, and also in the comparison of the running time, the LPA algorithm has the best results, thus indicating that among the three algorithms based on fuzzy subordinative degree, LPA is

**Table 3 Comparison of modularity.**

| Network | Time (ms) | | |
|---|---|---|---|
| | FSCA | KLA | LPA |
| Karate | 0.371 | 0.372 | 0.416 |
| Football | 0.510 | 0.396 | 0.526 |
| Dolphin | 0.557 | 0.400 | 0.605 |

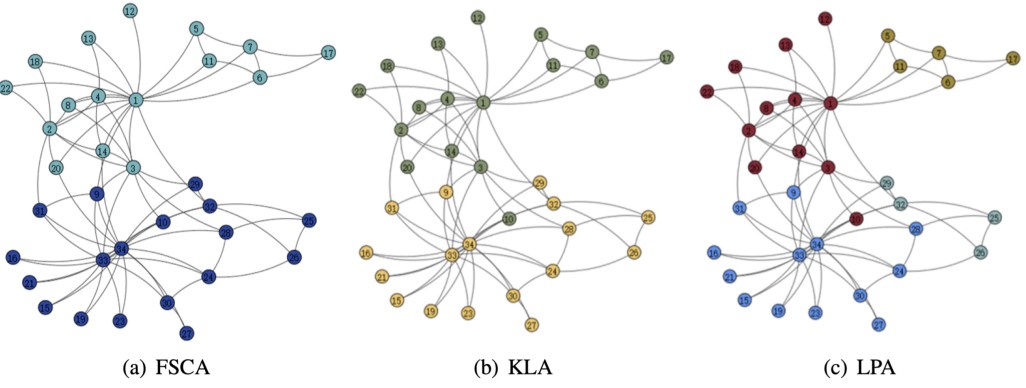

(a) FSCA                    (b) KLA                    (c) LPA

**Figure 5 Visualization of the karate.**

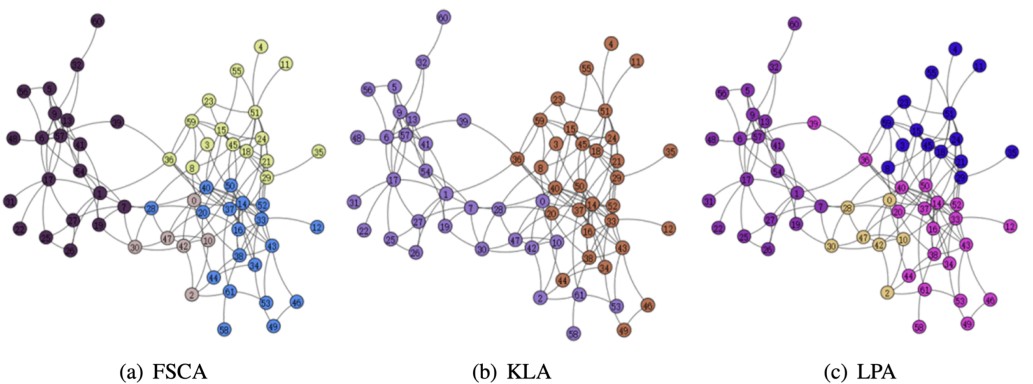

(a) FSCA                    (b) KLA                    (c) LPA

**Figure 6 Visualization of the dolphin.**

the best algorithm. The visualization of the three algorithms is shown in Figs. 5 and 6. Therefore, LPA will be used as the default community division algorithm in anonymity graph construction.

## Compare original k-anonymization

The original k-anonymity technique, when applied to graph data for privacy protection, often disrupts the community structure within the network because it requires k-anonymity to be enforced across the entire network. However, by initially dividing the network into communities and then applying k-anonymity protection, the integrity of the

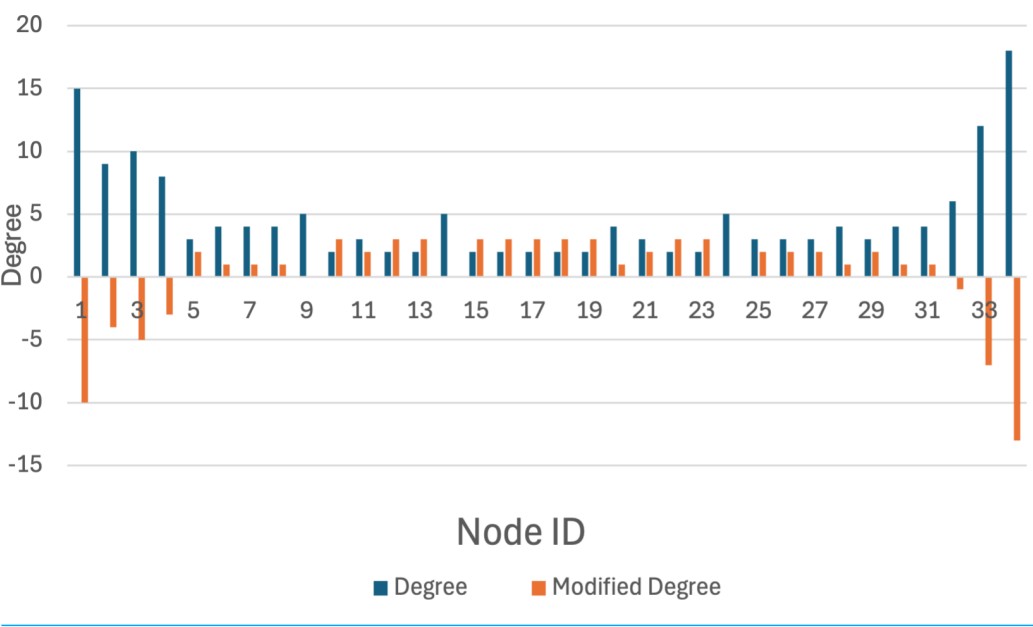

**Figure 7** Original k-anonymization algorithm.

original community structures can be maintained, which in turn enhances data usability. Furthermore, community detection allows for the implementation of varying levels of privacy protection tailored to the data owner's specific privacy requirements.

The karate network serves as an illustrative example to demonstrate that the privacy-preserving scheme proposed in this article achieves a balance between privacy preservation and data release. Initially, the original K-anonymization algorithm, which sets the average degree of the network as the $k$ value, is applied. The results are depicted in Fig. 7, where the degree change amounts to 94, while the number of edges that need to be modified is 47. Subsequently, by employing the K-anonymization algorithm after integrating community division, the network is initially divided into two communities using a community mining algorithm and assumes that the degree is greater than the threshold. As shown in Fig. 8, where C1 and C2 represent the communities in karate, the number of modified edges is significantly reduced compared to the original K-anonymization algorithm. Specifically, eight edges need modification for community 1, and 5 edges for community 2. This results in a total reduction of nearly 72% in the number of modified edges compared to the original algorithm.

## CONCLUSION

In this article, a social network privacy protection scheme based on anonymized graph construction is proposed. Firstly, leveraging community characteristics, the concept of fuzzy subordinative degree is introduced. Subsequently, three community structure mining algorithms are devised: the fuzzy subordinative degree-based algorithm, the improved KL algorithm, and the enhanced label propagation algorithm. Thirdly, various anonymity graph construction algorithms are designed based on community structure to

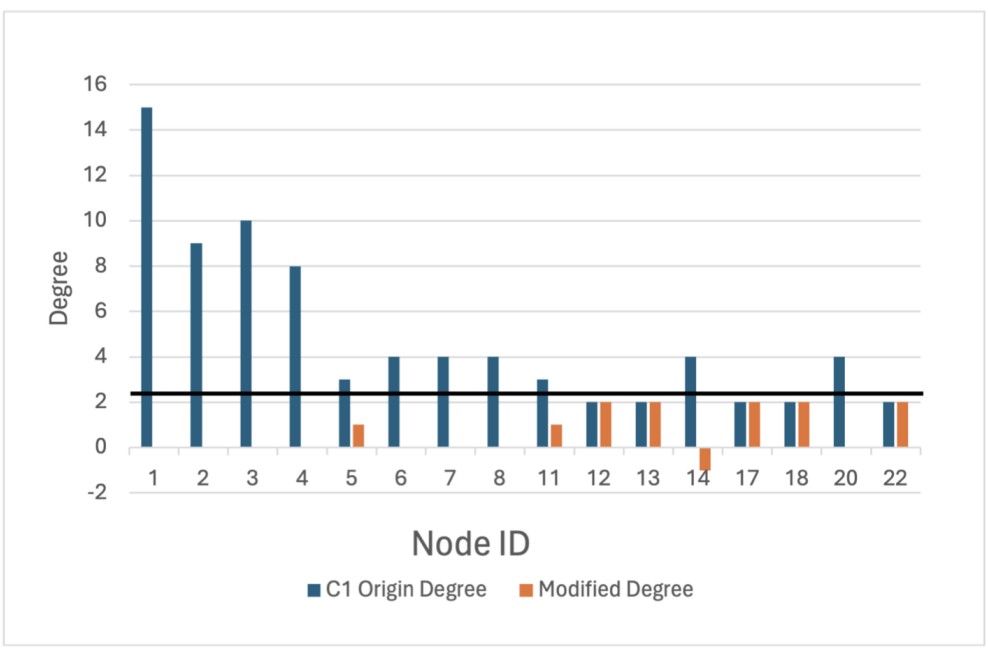

(a) community1 degree modify

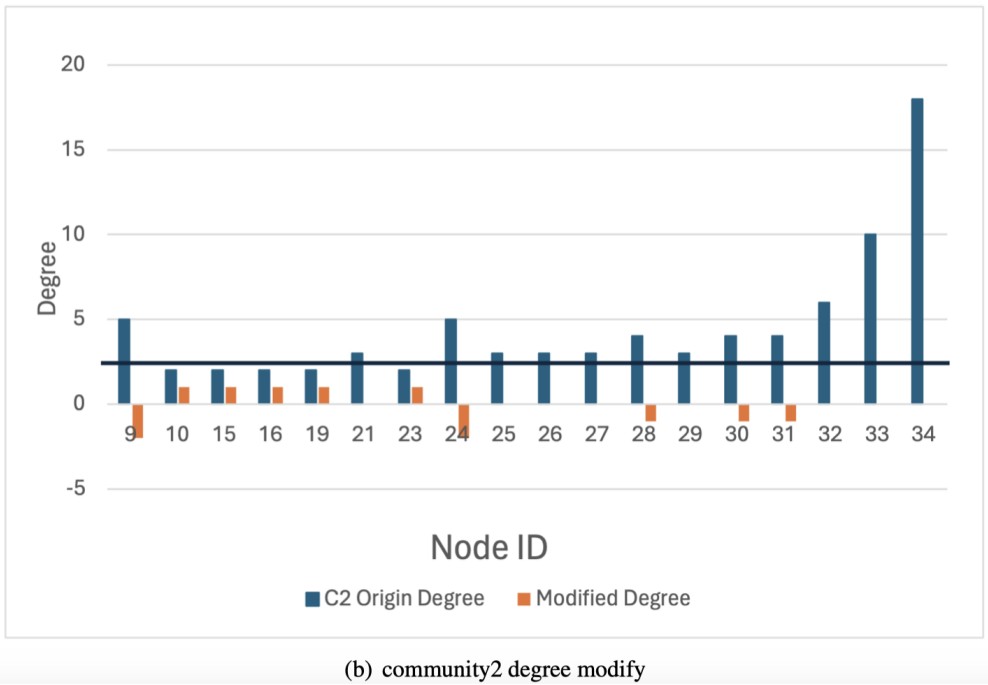

(b) community2 degree modify

**Figure 8**  **Anonymous graph modification based on community.**

accommodate different privacy protection levels. Simulation experiments demonstrate the effectiveness of the three methods in network partitioning across different network sizes. Additionally, the anonymized graph construction method based on community structure satisfactorily meets privacy requirements with fewer modifications.

This article presents a dynamically adjustable model for privacy preservation, suitable for networks of various sizes. However, the specific network size and privacy protection level currently require manual definition, with the next step aimed at automating this process. Additionally, there is room for further optimization in the construction of anonymity graphs. Given the increasing sophistication of attacks, such as inference attacks, it is imperative to design more robust anonymity graph construction methods capable of withstanding such threats.

### Funding

This research is supported by the National Natural Science Foundation of China (No. 61902069) and the Natural Science Foundation of Fujian Province of China (2021J011068). The funders had no role in study design, data collection and analysis, decision to publish, or preparation of the manuscript.

### Grant Disclosures

The following grant information was disclosed by the authors:
The National Natural Science Foundation of China: No. 61902069.
The Natural Science Foundation of Fujian Province of China: 2021J011068.

### Competing Interests

The authors declare there are no competing interests.

### Author Contributions

- Linghong Kuang conceived and designed the experiments, performed the experiments, analyzed the data, performed the computation work, prepared figures and/or tables, and approved the final draft.
- Kunliang Si performed the experiments, analyzed the data, performed the computation work, prepared figures and/or tables, and approved the final draft.
- Jing Zhang performed the experiments, analyzed the data, authored or reviewed drafts of the article, and approved the final draft.

### Data Availability

The raw data and code are available in the Supplemental Files.
The dolphins dataset is available from: https://doi.org/10.1007/s00265-003-0651-y.
The football dataset is available from: https://doi.org/10.1073/pnas.122653799.
The karate dataset is available from: https://doi.org/10.1086/jar.33.4.3629752.

## Supplemental Information

Supplemental information for this article can be found online at http://dx.doi.org/10.7717/peerj-cs.2244#supplemental-information.

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
