# Peer review of "Anonymous group structure algorithm based on community structure"

_PeerJ Computer Science, doi:10.7717/peerj-cs.2244_

## Round 0.1 · original submission · Major Revisions

The reviewers have given suggestions and comments. Please revise the article according to the comments. Then the paper will be evaluated again.

Reviewer 1 ·

Basic reporting

This manuscript focuses on protecting the privacy leakage of data sharing. The authors propose an anonymous group structure algorithm based on community structure. This work is of practical significance. I have the following comments:
(1) I hope the authors could add more updated references, especially in the recent years.
(2) The authors should pay more attention to the spelling. In the algorithm table, the abbreviation of the algorithm is NSA, is it right? And there are some spelling errors. Such as, on page 8, “Experiments on three public social network datasets (Dataset dowloand from xx)”, “Table 2 displays the runtime performance of the three algorithms in the fou dataset”.
(3) Authors should descript the mechanism of proposed algorithm. It would be better to add a figure about the algorithm framework.

Experimental design

no comment

Validity of the findings

no comment

Cite this review as

Reviewer 2 ·

Basic reporting

This paper proposed a novel privacy protection scheme for social networks based on Anonymous group structure. The authors discussed thoroughly their proposed algorithms, as well as related theoretical background and mathematical proofs.

Experimental design

Comprehensive experiments are included to demonstrate the performance of the proposed scheme.

Validity of the findings

However, I also found the following issues:
1. The name and abbreviation of Algorithm 1 are not consistent.
2. The layout of Algorithm 3 and Form 2 is not appropriate.
3. Formula 3 should be centered.
4. Definitions in PRELIMINARIES should give the source.

Additional comments

In summary, the research work in this article is valuable and that makes a valuable contribution to the utilization of anonymity technology in bolstering social network protection. After addressing the issues mentioned earlier, the article will present more comprehensive and lucid description of the work conducted.

Cite this review as

Reviewer 3 ·

Basic reporting

(1) Some formula symbols are not very clear and accurate. In line 146, r1 may have been used incorrectly. In line 152-155, n does not appear in formula 2, and di, dj ∑ij, Γ and so on are also not explained. In Algorithm 1, η2 is not set in advance.
(2) Several sentences have unclear meanings. In line 250, what is 'fou dataset' ? In Figure 7, 'Modify' should be replaced with ‘Modified'. In Figure 8, what is 'C1' or 'C2'? In line 281, ’ Secondly‘ should be 'Thirdly'?
(3) In Figure 7 and 8, the horizontal axis should be named. A line of the average degree can be drawn and prominently marked "modified degree" of each nodes in Figure 7.

Experimental design

(1) Algorithm 4 is not used in the experiment. And, there is no explanation on how to select a level value in the article.
(2) In 'Compare original K-anonymization', simulation result analysis focuses on the comparison of 'degree'. Considering the purpose of this article is to develop anonymous group structure Algorithm, suggest to discuss some certain characteristic indicators of ’anonymous group structure‘ with different network sizes.

Validity of the findings

(1) In ABSTRACT, 'They can be utilized at different network sizes' is proposed. But, no specific judgment rules or standards of network sizes were proposed after Experiments.
(2) In line 44-46, three 'how to' questions were proposed. But, only the second one is fully discussed in the article. The others also need to be reflected in the discussion methods and experiments.
(3) In line 278, 'In this paper, a data-sharing scheme based on anonymized graph construction is proposed.' But, the purpose of this paper is not to propose 'a data-sharing scheme', so, suuggest adjusting the description of the conclusion.

Cite this review as

---

## Round 0.2 · accepted · Accept

Thanks to the authors for your efforts to improve the work. This version successfully satisfied the reviewer. It can be accepted now.

Reviewer 3 ·

Basic reporting

no comment

Experimental design

The experimental results verify the effectiveness of community mining, but do not show that it can improve the effect of privacy protection. It is recommended to increase this analysis.

Validity of the findings

Authors put forward that the main work of the paper is to put forward the ' a privacy protection scheme '. However, the title of the paper does not reflect this point, and it is suggested to adjust it.

Additional comments

no comment

Cite this review as